# Mistakes can stabilise the dynamics of rock-paper-scissors games

**Maria Kleshnina**[1]*, **Sabrina S. Streipert**[2], **Jerzy A. Filar**[3], **Krishnendu Chatterjee**[1]

1 Institute of Science and Technology Austria (IST Austria), Klosterneuburg, Austria, 2 Department of Mathematics and Statistics, McMaster University, Hamilton, Canada, 3 School of Mathematics and Physics, University of Queensland, Brisbane, Australia

* maria.kleshnina@ist.ac.at

## Abstract

A game of rock-paper-scissors is an interesting example of an interaction where none of the pure strategies strictly dominates all others, leading to a cyclic pattern. In this work, we consider an unstable version of rock-paper-scissors dynamics and allow individuals to make behavioural mistakes during the strategy execution. We show that such an assumption can break a cyclic relationship leading to a stable equilibrium emerging with only one strategy surviving. We consider two cases: completely random mistakes when individuals have no bias towards any strategy and a general form of mistakes. Then, we determine conditions for a strategy to dominate all other strategies. However, given that individuals who adopt a dominating strategy are still prone to behavioural mistakes in the observed behaviour, we may still observe extinct strategies. That is, behavioural mistakes in strategy execution stabilise evolutionary dynamics leading to an evolutionary stable and, potentially, mixed co-existence equilibrium.

## Author summary

A game of rock-paper-scissors is more than just a children's game. This type of interactions is often used to describe competition among animals or humans. A special feature of such an interaction is that none of the pure strategies dominates, resulting in a cyclic pattern. However, in wild communities such interactions are rarely observed by biologists. Our results suggest that this lack of cyclicity may stem from imperfectness of interacting individuals. In other words, we show analytically that heterogeneity in behavioural patterns may break a cyclic relationship and lead to a stable equilibrium in pure or mixed strategies.

## Introduction

The question frequently arising in ecology is: *Under which conditions does a particular type of species survive?* This question is also relevant in the context of understanding a wide range of environmental, social, genetic and other conditions potentially influencing evolutionary

**Data Availability Statement:** All relevant data are within the manuscript and its Supporting information files.

**Funding:** This study was supported by the following grants: the European Union's Horizon

2020 research and innovation program under the Marie Sklodowska-Curie Grant Agreement #754411, to MK; the Australian Research Council Discovery Grant DP180101602, to JAF; the European Research Council Consolidator Grant 863818 (FoRM-SMArt) to KC. The funders had no role in study design, data collection and analysis, decision to publish, or preparation of the manuscript.

**Competing interests:** The authors have declared that no competing interests exist.

trajectories. Evolutionary game theory, a branch of game theory and ecological sciences, aims to answer that question [1–5]. One of the most well-known games applied to biology is the rock-paper-scissors game (RPS). Here, rock beats scissors, scissors beat paper and paper beats rock. Whether we are talking about population dynamics or economics and human behaviour, this game is known to illustrate salient features while being easy to understand (for a thorough review of the models used to study RPS games see [6]). In biology, this game was applied to explain cyclic dynamics in some species such as mating strategies of side-blotched lizards [7, 8] and phenotypic competition in bacterial strains of *E. Coli* [9, 10]. Furthermore, in the engineered microbial populations, introduction of such a competition seemed to stabilise the community [11] and even promote cooperation [12]. Moreover, it was suggested that introduction of new strategies into classic social dilemmas, such as loners [13–15] or risk-averse hedgers [16], can lead to cyclic competition. Nevertheless, cyclicity is rarely observed in wild communities of microbes [17], even though it was shown experimentally that behavioural heterogeneity in microbes can stabilise communities [18]. Recently, it was suggested that it might be challenging for such non-transitive competition to evolve in the first place [19]. However, even if cyclic competition emerges, its stability can be very sensitive to the exact balance in the community, potentially leading to the dominance of only one strategy [20]. In this paper, we utilise a game-theoretic concept of incompetence [21, 22] which allows individuals to make mistakes during the execution of their strategy. This results in a potentially unintended strategy being actually played during the interaction with another individual. We show that such an assumption can induce evolutionary stability in the initially unstable rock-paper-scissors dynamics and predict possible outcomes of the competition under the assumption of execution errors.

Behavioural stochasticity is an expanding field rich in different approaches to the problem. An approximation of behavioural errors of players in games was first considered as "trembling hands" [23] with the presence of mistakes during the strategies' execution with some small probability. Later, in evolutionary games it was modelled via mutations [24, 25], language learning [26–28] or other experimental learning processes [29–32], adaptation dynamics [33], phenotypic plasticity [34], edge diversity in games on graphs [35, 36], and noise in continuous and discrete-time replicator dynamics [37–40]. Furthermore, mutations of players were introduced to the replicator dynamics via the replicator-mutator dynamics [28, 41], where each type has its own mutation rate but these mutations do not occur simultaneously. However, behavioural stochasticity at the moment of interaction was not considered in these studies.

An attempt to generalise players' behavioural mistakes via the notion of incompetence was made in classic game theory [42]. Later, the concept of evolutionary games under incompetence was suggested to model such social problems of species in biological settings [22]. The notion of incompetence proposes a general framework for modelling behavioural mistakes with the underlying assumption that only one of the $n$ non-cooperative strategies can be executed. That is, with a certain probability, individuals might execute a strategy different from the one they chose. In these settings, both players are prone to making mistakes resulting in stochastic payoffs of all involved individuals, altering overall population's fitness.

Here, we consider the following scenario. Imagine, each randomly chosen individual finds itself in the pairwise interaction with another randomly chosen individual. Both of them choose a strategy to play. However, the chance that they will play their chosen strategies depends on two factors: on the overall level of behavioural plasticity in the population and a distribution of behavioural mistakes. If the population is completely homogeneous, then all interactions among the individuals are deterministic ($\lambda = 1$, see Fig 1A). However, if the population's behaviour is plastic ($\lambda < 1$), then individuals may make mistakes when executing their chosen strategies. The probabilities of playing one or another strategy are determined both by the degree of plasticity, $\lambda$, and their maximal probabilities of mistakes captured in matrix

**Fig 1. A schematic representation of behavioural mistakes in a rock-paper-scissors game.** (A) Rock-paper-scissors dynamics with pure strategies is described by a fitness matrix such that the cyclic relationship between the three strategies is promoted. (B) The effect of execution errors on the example of one interaction: here individual 1 has chosen strategy paper and individual 2 has chosen strategy rock. Without mistakes, individual 1 would win this instance of the contest. However, a mistake in the execution leads to mixed strategies being played for both individuals resulting in different possible outcomes of the interaction. Hence, the outcome of the game is no longer deterministic but stochastic and depends on the probability distribution of mistakes.

$S$ ($\lambda = 0$). The latter results in behavioural plasticity that perturbs the game outcome (see Fig 1B). In some games, execution errors mean that organisms are able to execute strategies required by the environmental conditions even when they make a wrong choice. That is, species execute strategies that are required for their survival in the environment, by mistake. We do not assume that they carry out this execution consciously. However, this random characteristic may be crucial when we consider changing environments where adaptation becomes particularly important and depends strongly on the interplay between behavioural patterns and fitness.

In low-dimensional games this interplay can be captured and analysed in detail. Unfortunately, it becomes challenging as dimensionality of a game grows where even small perturbations may impact an evolutionary outcome. However, under a natural assumption that behavioural mistakes are completely random, we can describe game behaviour for general $n$ dimensions. We show that in such settings, strategies (or behavioural types) leverage their fitness advantage. This in turn might lead to only one strategy dominating. Further, we assume that mistakes do not have to be completely random. We consider a symmetric case of an unstable RPS game where no choice of strategies yields a fitness advantage. Such games lead to a heteroclinic orbit where none of the strategies dominate. We choose such settings precisely because it is challenging to induce stability in these games. By contrast, an initially stable version of the RPS game can promote biodiversity even in finite populations settings [43], and even very small perturbations can stabilise a classic version of the RPS game [44]. We show that behavioural mistakes bring asymmetry to the game, breaking the cyclic relationship and potentially leading to dominance of one of the strategies. That is, the structure of execution errors may technically imply the existence of an evolutionary stable interior point.

## Model

In this paper we focus on the RPS dynamics. Hence, we shall mostly work with the general form of $R$ given by

$$
\begin{array}{c}
\begin{array}{ccc} \textit{Rock} & \textit{Paper} & \textit{Scissors} \end{array} \\
\begin{array}{c} \textit{Rock} \\ \textit{Paper} \\ \textit{Scissors} \end{array}
\begin{pmatrix}
0 & -a_1 & b_1 \\
b_2 & 0 & -a_2 \\
-a_3 & b_3 & 0
\end{pmatrix},
\end{array}
\tag{1}
$$

where $a_i, b_i \in \mathbb{R}^+$ [4].

In classic games, there is an underlying assumption that players are able to execute the chosen actions perfectly. We assume that actions selected by players may not coincide with the executed actions. Such behavioural stochasticity results in executing unintended strategies and is captured in matrix $Q(\lambda)$ from [21] defined as

$$Q(\lambda) = (1 - \lambda)S + \lambda I, \ \lambda \in [0, 1]. \tag{2}$$

In [21] the authors called $Q(\lambda)$ the incompetence matrix with elements $q_{ij}(\lambda)$. However, in the biological context considered here the name *plasticity matrix* is more appropriate. This stochastic matrix is constructed from the set of all probabilities of player 1 executing action $j$ given that she selects action $i$. When $\lambda = 1$, $Q(1) = I$ and no mistakes are observed in the population. Hence, the population is behaviourally homogeneous and all interactions are deterministic. However, if $\lambda < 1$, then with probabilities $q_{ij}(\lambda)$ an individual chooses to play strategy $i$ but plays strategy $j$ instead. We say that in such a case the population is $Q(\lambda)$-heterogeneous and the outcomes of the interactions are now stochastic. We shall call $\lambda$ the *strength of behavioural plasticity*. In the limit as $\lambda \to 0$, the matrix $Q(0)$ is equal to $S$, which is defined as a limiting distribution of behavioural mistakes. Such a matrix in the case of a three-strategy matrix game has the form

$$S = \begin{pmatrix} s_{11} & s_{12} & s_{13} \\ s_{21} & s_{22} & s_{23} \\ s_{31} & s_{32} & s_{33} \end{pmatrix}, \tag{3}$$

and is also a stochastic matrix. Every $i$-th row of this matrix defines a mixed profile of each strategy $i$. We define the expected incompetent reward matrix as a perturbation of the fitness matrix by plasticity (or incompetence), namely

$$R(\lambda) = Q(\lambda)RQ(\lambda)^T. \tag{4}$$

It is sufficient to consider the following simpler canonical form of the fitness matrix

$$\tilde{R}(\lambda) = R(\lambda) - D(R(\lambda)), \tag{5}$$

where $D(R(\lambda))$ is a matrix with each column $j$ consisting of the diagonal elements of $R(\lambda)$, inducing $r_{jj}(\lambda) = 0$, $j = 1, 2, 3$ since such positive linear transformation of the fitness matrix does not affect the qualitative behaviour of replicator dynamics [45]. In our further analysis, we will focus on the equilibrium analysis of the games with the fitness matrix $\tilde{R}(\lambda)$, and explore possible transitions caused by $\lambda$ changing values in [0, 1]. Then, substituting (4) into (5), we obtain a new game with mistakes $\tilde{R}(\lambda)$ given by

$$\tilde{R}(\lambda) = \begin{pmatrix} 0 & \tilde{r}_{12} & \tilde{r}_{13} \\ \tilde{r}_{21} & 0 & \tilde{r}_{23} \\ \tilde{r}_{31} & \tilde{r}_{32} & 0 \end{pmatrix}, \tag{6}$$

where every element of the fitness matrix $\tilde{R}(\lambda)$ has the form

$$\tilde{r}_{ij}(\lambda) = (\mathbf{q}_i - \mathbf{q}_j)^T R \mathbf{q}_j,$$

with $\mathbf{q}_i$ being the $i$-th row of matrix $Q(\lambda)$ from (2).

In the evolutionary sense, behavioural mistakes lead to perturbations in fitness that populations obtain over time. This might be due to populations' migration to new and unexplored environments or due to changing environments. Then, interacting individuals obtain a finite number, $n$, of available behavioural strategies. With the absence of mistakes, both interacting individuals are making their strategical choices which lead to some payoff according to the fitness matrix $R$. However, mistakes from matrix $Q(\lambda)$ perturb the outcome of the interaction twice as both interacting individuals are prone to execution errors. Hence, the population dynamics now depends on the degree of plasticity, that is competency of individuals, according to replicator equations [46] defined as

$$\dot{x}_i = x_i(f_i(\lambda) - \phi(\lambda)), \quad i = 1, \dots, n,$$

where the fitness of $i$-th strategy is given by

$$f_i(\lambda) = \mathbf{e}_i^T \tilde{R}(\lambda)\mathbf{x}$$

where $\mathbf{e}_i^T$ is the $i$-th transposed unit basis vector. The mean fitness of the entire population is defined as

$$\phi(\lambda) = \mathbf{x}^T \tilde{R}(\lambda)\mathbf{x}.$$

## Interpretation of $\lambda$

The model proposed here was first referred to as a "game with incompetence of players" [21, 42]. That is, the matrix $Q$ was consisting of probabilities of players' mistakes, when they intended to execute strategy $i$ but played strategy $j$ instead. Such a model was inspired by an analogy with tennis players, where less experienced players are prone to hitting a different shot to one they initially intended. Here, players have a set of $n$ possible shots to hit. Given the complexity level of the shot as well as players' talents, those probabilities of mistakes will not be uniform. Moreover, players are learning while training and, hence, reducing their incompetence. This was captured in the parameter $\lambda$: with the level of mistakes decreasing as $\lambda \to 1$.

This concept was next considered in the evolutionary settings as a modelling approach to adaptation to a new environment [22]. First, it was assumed that a population is immersed into a new environment, which can happen either due to migration of animals or changing environmental conditions. It is assumed that there are $n$ behavioural types or strategies available to individuals. Then, new conditions might increase stress levels and force individuals' behaviour to deviate from the one in the old environment. Such deviations are then captured in the matrix $S$. As time passes by, animals learn and adapt to their new environmental conditions, which is then reflected in the parameter $\lambda$. In such settings, one can also assume some form of learning dynamics, $\lambda(t)$ [47].

Another possible way to think about this model, is to apply it at a genetic level [48]. That is, we would construct a game between $n$ pure types, for instance, genes in microbes. The time-dependent process of $\lambda(t)$ evolving from 1 to 0 can then be considered more as environmental stimuli dynamics and have various functional forms reflecting environmental fluctuations. Matrices $S$ and $Q$ would represent levels of phenotypic plasticity, where each phenotype would allow some mixing between $n$ genes that depend on the level of environmental stimuli. Then, natural selection would drive the evolution, which might result in extinction of one type or another. This also depends on the assumption concerning the exact form of environmental fluctuations.

Here, we focus on the more general interpretation of $\lambda$ as the strength of behavioural plasticity. For this general approach we do not impose any time-dependence on $\lambda$. Instead, we

study all possible equilibria for each of the values of λ in the interval [0, 1]. Every pure strategy *i* has an assigned probability distribution captured in the matrix $Q(\lambda)$. When λ = 0, the population utilises a limiting distribution of mistakes *S* and has maximal plasticity. When λ = 1, the population's behaviour is deterministic and no plasticity is observed. This can be interpreted as an approach to modelling behavioural heterogeneity or noise in interactions. Specifically, in the settings of phenotypic plasticity, it is natural to assume a complete randomisation in the strategy execution corresponding to *S* being comprised of uniformly distributed probability vectors. However, in terms of adaptations to new environmental conditions, probability of mistakes may differ depending on the strategy being chosen. Thus, we shall assume a general form of matrix *S*. Next, we shall first demonstrate this model on some examples.

### Motivating example 1

First, consider phenotypic behavioural plasticity as an interpretation of the model. In such settings, it is natural to assume that "execution errors" are symmetric and equally likely. That is, let us assume that if λ = 0, then individuals are completely random in their strategic choice. Then, all components of matrix *S* are equal and are given by $s_{ij} = {}^{1}/_{n}$. Next, let us assume that strategies are not equal in their fitness advantages by considering the fitness matrix *R* given by

$$\begin{array}{c} & \begin{array}{ccc} Rock & Paper & Scissors \end{array} \\ \begin{array}{c} Rock \\ Paper \\ Scissors \end{array} & \begin{pmatrix} 0 & -2 & 1 \\ 1 & 0 & -3 \\ -3 & 1 & 0 \end{pmatrix} \end{array}.$$

Game flows for different values of λ are depicted in Fig 2. For λ = 1, the game possesses an unstable mixed equilibrium $\tilde{\mathbf{x}} = \left({}^{10}/_{32}, {}^{13}/_{32}, {}^{9}/_{32}\right)$ (see Fig 2A). As the strength of behavioural plasticity increases (λ decreases), the game dynamics experiences several transitions. First, the interior equilibrium of the game with pure strategies is pushed to the population adopting only rock strategy (Fig 2B and 2C) via the existence of the unstable equilibrium point on the paper-rock edge. Note that in panel B an interior equilibrium point still exists whereas in panel C the game transits to having no interior equilibrium.

Since the stable equilibrium is a strict Nash equilibrium, it is an evolutionary stable strategy (ESS) [4]. However, for any given λ and strategy choice, $\tilde{\mathbf{x}}(\lambda)$, the observed stochastic behaviour of organisms, $\tilde{\mathbf{y}}(\lambda)$, is defined by the matrix $Q(\lambda)$ as a result of

$$\tilde{\mathbf{y}}(\lambda) = Q(\lambda)\tilde{\mathbf{x}}(\lambda). \tag{7}$$

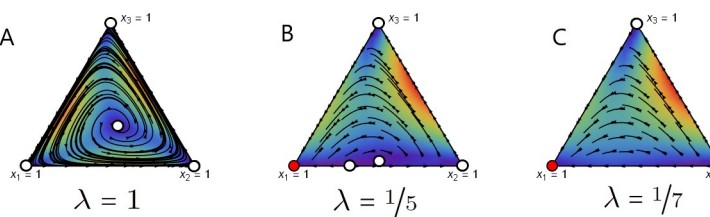

**Fig 2. Game flow for the unstable RPS game with uniform mixed strategies for different values of λ.** Here, a stable fixed point is denoted by a red circle and a unstable fixed point is denoted by a white circle. The colour in the interior of the simplex indicates the rate of change: from slow (blue) to fast (red). In this example, completely random execution errors lead to the dominance of the rock strategy. We use the Wolfram Mathematica project [49] to produce these phase planes.

Hence, at $\lambda = 0$ the game possesses a stable pure equilibrium $\tilde{\mathbf{x}}(0) = (1, 0, 0)$ that corresponds to the execution vector $\tilde{\mathbf{y}} = \mathbf{s}_1^T = (^1/_3, ^1/_3, ^1/_3)$, where $\mathbf{s}_1^T$ is the first column of the matrix $S$. That is, if $\lambda$ is sufficiently close to 0, the game $\tilde{R}(0)$ obtains a stable interior completely mixed equilibrium. Hence, given the execution vector, the assumption of behavioural plasticity introduces a stable interior ESS to an unstable rock-paper-scissors game. Note that, from the perspective of the observed strategy, it does not matter which of the strategies will dominate in this case as they all have the same probabilities of mistakes.

## Motivating example 2

The assumption that individuals make mistakes completely at random is somewhat limiting. In some cases, more freedom in the definition of individuals' plasticity is required. For instance, if we assume that $\lambda$ is interpreted as an adaptation process to new environmental conditions, then some behavioural choices may have different distributions of mistakes. For instance, let us consider an example where the fitness matrix $R$ is given as follows

$$
\begin{array}{cccc}
 & Rock & Paper & Scissors \\
Rock & \begin{pmatrix} 0 \\ \\ 1 \\ \\ -2 \end{pmatrix} & \begin{matrix} -2 \\ \\ 0 \\ \\ 1 \end{matrix} & \begin{matrix} 1 \\ \\ -2 \\ \\ 0 \end{matrix} \end{array}.
$$

Note that the determinant of $R$ is negative, which implies that this game possesses an interior fixed point $(^1/_3, ^1/_3, ^1/_3)$ that is unstable. Hence, there exists a heteroclinic orbit as there are no stable equilibria. The dynamics then oscillate from the centre of the simplex to the boundaries. Such dynamics are generally quite robust under perturbations. We now consider how the game dynamics behave under our assumptions.

The exact probability distributions captured in $S$ would depend on the particular situation and species under consideration. Let us demonstrate the influence of execution errors on the following example of matrix $S$. Assume that at the highest level of execution errors ($\lambda = 0$) individuals play each of their chosen strategies with probability not less than $^1/_3$. When an individual plays a scissors strategy, her strategy execution is completely random. However, choosing a rock or paper strategy may induce some asymmetry in the strategy execution. An individual playing a rock strategy executes only rock and paper strategies with probabilities $^1/_3$ and $^2/_3$, respectively. Individuals, who choose a paper strategy, obtain a limiting distribution of mistakes of $(^1/_2, ^1/_3, ^1/_6)$. Then, the matrix $S$ is given by

$$
\begin{array}{cccc}
 & Rock & Paper & Scissors \\
Rock & \begin{pmatrix} ^1/_3 \\ \\ ^1/_2 \\ \\ ^1/_3 \end{pmatrix} & \begin{matrix} ^2/_3 \\ \\ ^1/_3 \\ \\ ^1/_3 \end{matrix} & \begin{matrix} 0 \\ \\ ^1/_6 \\ \\ ^1/_3 \end{matrix} \end{array}.
$$

Game flows for different values of $\lambda$ are depicted in Fig 3. As $\lambda$ varies from 1 to 0, the game dynamics go through several transitions (see panel A for the overview). The first transition happens at $\lambda = ^2/_5$ when pure stable equilibrium of scissors emerges (see Fig 3B and 3C). Note that the interior equilibrium still exists but the heteroclinic orbit does not—the dynamics converge to a stable point. Next, at $\lambda \approx 0.287$, a paper strategy becomes stable (Fig 3D). Further, the interior fixed point vanishes at $\lambda = ^1/_4$ leaving unstable fixed points on the rock-

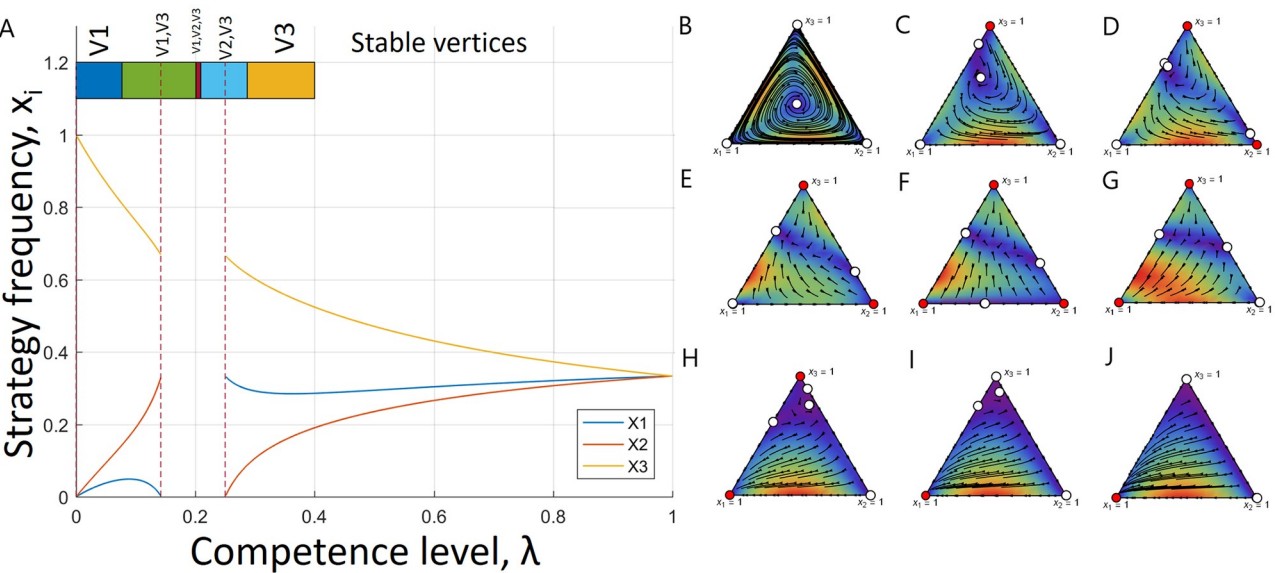

**Fig 3. Game transitions under execution errors from example 2.** (A) Frequencies of each strategies in the interior equilibrium as functions of $\lambda$. Here, $x_1$ represents rock frequency, $x_2$—paper frequency and $x_3$—scissors frequency. The interior equilibrium exists for most the values of $\lambda$ but $(^1/_7, ^1/_4)$. Further, the coloured bar at the top of the plot indicates stability intervals of $\lambda$ for different vertices (a stable vertex is indicated on top of the bar). For instance, vertex 3 is the only stable vertex for $\lambda \in (\approx 0.287, ^2/_5)$. Game flow for the unstable rock-paper-scissors game is depicted for different values of $\lambda$ as follows: (B) $\lambda = 1$, (C) $\lambda = 0.3$, (D) $\lambda = 0.27$, (E) $\lambda = 0.22$, (F) $\lambda = 0.205$, (G) $\lambda = 0.16$, (H) $\lambda = 0.1$, (I) $\lambda = 0.05$, (J) $\lambda = 0$. We depicted each transition in the game from panel A. Here, a stable fixed point is denoted by a red circle and a unstable fixed point is denoted by a white circle. Hence, as $\lambda$ changes its values from 1 to 0, the game experiences several transitions in its equilibria and for different degrees of execution errors, each of the pure strategies has a chance to dominate. However, for the maximum plasticity, only pure rock strategy survives.

scissors and paper-scissors edges (Fig 3E), which is followed by a rock strategy becoming stable at $\lambda \approx 0.209$ (Fig 3F). This interval of all three strategies being stable is rather short as at $\lambda = ^1/_5$ paper loses its stability (Fig 3G). However, these are not the only transformations occurring: at $\lambda = ^1/_7$ the interior equilibrium emerges again (Fig 3H). While the interior equilibrium exists again, scissors lose their stability too at $\lambda = ^1/_{13}$ and at $\lambda = 0$ rock is the only stable equilibrium. Note that for $\lambda = 0$ (Fig 3J), the stable observed pure equilibrium will again be a mixed strategy due to the execution errors given by $\tilde{\mathbf{y}}(0) = \mathbf{s}_1 = (^1/_3, ^2/_3, 0)$.

These examples demonstrate that execution errors might break the heteroclinic orbit by introducing a stable equilibrium in the game. That is, stochasticity induced by mistakes might stabilise dynamics that were unstable before. In addition, in the case of players executing only mixed strategies, the game might obtain a stable interior point altering its original dynamics (see Figs 2C and 3J). In the following analysis we shall examine possible transitions in unstable RPS games. We aim to define conditions under which we can secure existence of a stable equilibrium.

## Results

### Games with completely random plasticity

Let us first consider the case when behavioural mistakes are completely random. Such settings can be interpreted as either a form of phenotypic plasticity or just noise in the interactions. Then, the matrix $S$ is such that any strategy obtains the same probability of mistakes, that is, $s_{ij} = ^1/_3, \forall i, j = 1, 2, 3$. For such a game, the canonical fitness matrix simplifies to

$$\hat{R}(\lambda) = R + \frac{1-\lambda}{3\lambda}RJ,$$

where $J$ is a matrix of ones, $R$ is the fitness matrix of the original game and $\lambda \in (0, 1]$. In a game with $\lambda \in (0, 1]$, if strategies do not induce any overall fitness advantage to any strategy (that is, $R$ is a row-sum-constant matrix), then uniform execution errors will not affect the resulting equilibrium (see S1 File, Proposition 1).

**Result 1**. *Let $\tilde{\mathbf{x}}$ be an interior equilibrium of $R$. If the limiting distribution of mistakes, $S$, is a uniform matrix, that is, $s_{ij} = {}^1/_3, \forall i, j = 1, 2, 3$ and $R$ is a row-sum-constant matrix, then $\tilde{\mathbf{x}}$ is an interior equilibrium for the game $\hat{R}(\lambda)$ for any $\lambda \in (0, 1]$.*

In other words, if in a row-sum-constant game everyone is making mistakes with the same probabilities, then population dynamics are invariant under these mistakes. However, diversity in fitness advantages between the strategies might help one of the groups to benefit from behavioural heterogeneity of the population by leveraging its fitness advantage. We can calculate the interior fixed point in a general row-sum case as follows:

**Result 2**. *Let $\tilde{\mathbf{x}}$ be an interior fixed point of the original game $R$. Then, for $\lambda$ sufficiently close to 1 and the limiting distribution of mistakes, $S$, being a uniform matrix, that is, $s_{ij} = {}^1/_3, \forall i, j = 1, 2, 3$, we obtain that*

$$\tilde{\mathbf{x}}(\lambda) = \frac{1}{\lambda}\left( \tilde{\mathbf{x}} - \frac{1-\lambda}{3}\mathbf{1} \right), \tag{8}$$

*is an interior fixed point for the game $\hat{R}(\lambda)$.*

Note that, the point $\tilde{\mathbf{x}}(\lambda)$ from Eq (8) remains a fixed point of the replicator dynamics as long as it is preserved in the interior of the simplex, that is, as long as $\tilde{x}_k > {}^{(1-\lambda)}/_3, \forall k$ (see S1 File, Theorem 1). Hence, it can be easily verified that for this point to remain in the interior of the simplex for all $\lambda \in [0, 1]$ we must have that $\tilde{x}_k = {}^1/_3, \forall k$. The exact position of $\tilde{\mathbf{x}}$ is determined by the entries of the matrix $R$. Since for a general form of $R$ the interior equilibrium will not be located in the exact centre of the simplex, some strategies can go extinct first. This confirms that not only the strength of behavioural plasticity and probabilities of mistakes are important, but also the relative fitness advantages captured in the fitness matrix $R$. That is, mistakes can give a chance to some strategies to make use of their fitness advantage leading to the dominance of a particular strategy.

**Remark.** Note that Result 2 holds for any number of strategies $n$ and any game. For a general form of the result see S1 File.

Result 2 implies that the interior equilibrium of the original game is shifted by behavioural heterogeneity and drives less fit strategies to extinction. However, the observed strategy will remain the same for any dominating pure strategy due to the symmetry in mistakes distributions. Hence, uniform $S$ introduces evolutionary stability in the games with heteroclinic cycles. Moreover, for the extreme case of behavioural plasticity ($\lambda \approx 0$), this equilibrium will be close to a completely mixed equilibrium $({}^1/_3, {}^1/_3, {}^1/_3)$. Note that in the case of $\lambda = 0$, the matrix $\tilde{R}(0)$ is a zero matrix, meaning that the strategies are neutral and any point in the simplex is stable.

## Breaking the cyclic relationship

Next we address the question: What if behavioural mistakes of individuals are not necessarily uniformly distributed? For instance, if we treat the parameter $\lambda$ as some form of adaptation or learning, then the probabilities of mistakes might be different for different strategies. In such a case, we consider the general form of matrix $S$ as in Eq (3). In order to study the effect of the limiting distribution of mistakes (as $\lambda \to 0$), we shall focus on the form of a RPS game, where no strategy gains a fitness advantage. That is, we assume a row-sum-constant fitness matrix with an unstable equilibrium in the centre of the simplex by letting $a_1 = a_2 = a_3 = a$ and

$b_1 = b_2 = b_3 = b$ in the matrix (1). The condition $a > b$ ensures instability of the interior fixed point $(^1/_3, ^1/_3, ^1/_3)$ and, hence, the existence of a heteroclinic orbit.

In three dimensions (see (6)), transitions in a game are caused by either the elements $\tilde{r}_{ij}(\lambda)$ changing the sign, or cofactors $\tilde{R}_{ij}(\lambda)$ changing the sign, or the determinant of the fitness matrix $\det(\tilde{R}(\lambda))$ changing its sign [22, 50]. In the case of RPS games, the stability of the interior equilibria is determined by the sign of the determinant of the fitness matrix [51]. There are three cases: (a) if $\det(R) < 0$, then such a game obtains an unstable interior equilibrium resulting in a heteroclinic cycle; (b) if $\det(R) > 0$, then such an equilibrium is a stable fixed point; (c) if $\det(R) = 0$, then there exists an unstable interior equilibrium and periodic orbits. Then, under the assumptions of our model, the interior point's stability could potentially switch. That is, if the determinant of the fitness matrix $\tilde{R}(\lambda)$ changes its sign while the game is still a RPS game, the unstable interior point could become stable. However, the equilibrium behaviour of the game with $\tilde{R}(\lambda)$ is the same as $R(\lambda)$ by [45] given the relation between the two matrices in Eq (5). Since the determinant of the fitness matrix $R(\lambda)$ always preserves the same sign as $\det(R)$, then $\det(\tilde{R}(\lambda))$ also cannot change its sign while the interior point exists. Hence, stability properties of the interior equilibrium cannot be changed in our model. However, the equilibrium can be pushed to the boundary of the strategy space due to the asymmetry in the matrix $S$.

Note that for a homogeneous population ($\lambda = 1$) the interior fixed point, $\tilde{\mathbf{x}}$, can be calculated as

$$\tilde{x}_i = \frac{\sum_{j=1}^{3} R_{ji}}{\sum_{j=1}^{3} \sum_{k=1}^{3} R_{jk}}, \tag{9}$$

where $R_{ji}$ are cofactors of the matrix $R$. Then, as rate of execution errors of players increase ($\lambda \to 0$), the interior point $\tilde{\mathbf{x}}(\lambda)$ might transform and become infeasible as one or two of the components reach 0, that is, $\tilde{x}_i(\lambda) \to 0$ for some $i$.

Hence, depending on the probabilities of mistakes, we can describe possible transitions in the game dynamics induced by the changes in the strength of behavioural plasticity, $\lambda$. For instance, the game might possess an unstable interior equilibrium for any $\lambda$. However, the stability of the vertices will be disturbed as the entries of $\tilde{R}(\lambda)$ change their signs. An example of such a case can be found in Fig 4A, where the components of the interior equilibrium $\tilde{x}_i(\lambda)$ are plotted. The coloured bar at the top of each plot indicates the interval of $\lambda$ where the vertices (either one or two or all three) are stable. Further, Fig 4B shows an example of the game transitions with the interior equilibrium disappearing after some $\lambda$ and varying stability of the vertices is depicted. However, given the rational functional form of $\tilde{x}_i(\lambda)$, it is possible that the interior equilibrium exists for more than one sub-interval of $\lambda$. For instance, in Fig 4C and panel D the interior fixed point emerges twice as $\lambda$ changes from 0 to 1. Furthermore, stability transitions of the vertices in such cases are rich in structure. This is especially the case for the example depicted in Fig 4D.

Generally, components of the interior equilibrium are rational functions with numerators and denominators being 4-th order polynomials in $\lambda$. Consequently, there is a variety of possible behaviours. However, we can determine strict conditions for a vertex to be stable, based on its behaviour for a mixed strategy profile captured in the corresponding rows of the matrix $S$. Specifically, we can determine those conditions in the following result (see S1 File for more details).

**Result 3.** *Let* $\lambda^c \in (0, 1)$ *be such that* $\lambda^c = \min(\lambda_{kj}^c, \lambda_{ij}^c)$, *where* $\tilde{r}_{kj}(\lambda_{kj}^c) = 0$ *and* $\tilde{r}_{ij}(\lambda_{ij}^c) = 0$, *i, j, k are all distinct. Vertex j is a stable point of the replicator dynamics under incompetence for*

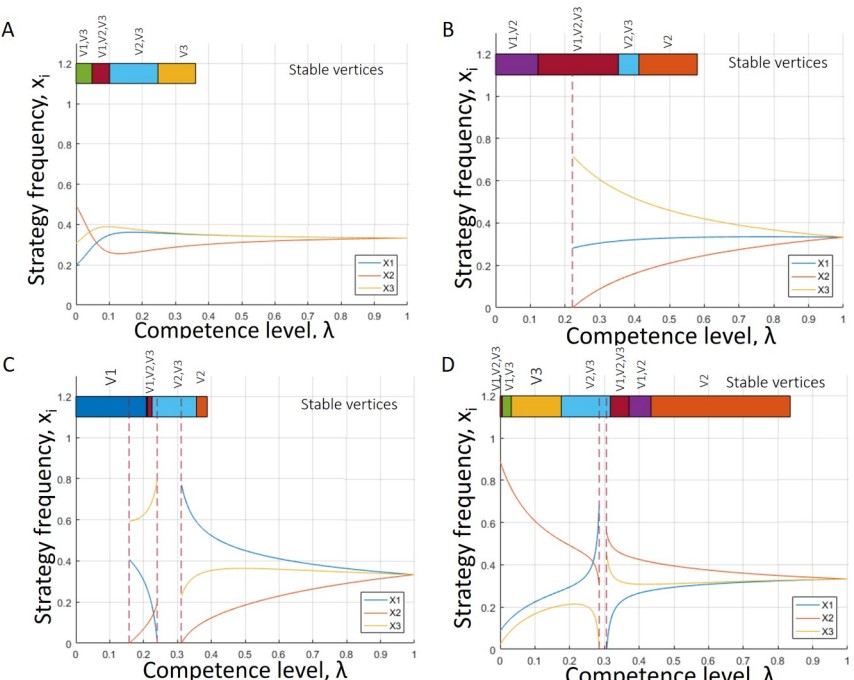

**Fig 4. Various game transitions of the unstable RPS game as λ varies between [0, 1].** The components of the interior fixed point are plotted as functions of λ. Further, the coloured bar at the top of the plot indicates stability intervals of λ for different vertices (a stable vertex is indicated on top of the bar). (A) The interior fixed point exists for all λ but vertices interchange their stability. In the limit of mistakes (λ → 0), two vertices are stable. (B) The interior fixed point exists for a sub-interval and vertices interchange their stability. As λ → 0, two vertices are stable. (C) The interior fixed point exists for two sub-intervals of (0, 1). In the limit of mistakes (λ → 0), only vertex 1 is stable. (D) The interior fixed point exists for almost all values of λ. In the limit of mistakes (λ → 0), all three vertices are stable. Generally, the exact equilibria transitions and existence of an interior equilibrium is determined by the limiting distribution of mistakes, *S*. We found that for almost all matrices *S* there is a high chance that at least one of the pure strategies will become dominant.

$\lambda \in [0, \lambda^c)$ *if and only if*

$$\mathbf{s}_j R \mathbf{s}_j > \mathbf{s}_l R \mathbf{s}_j, \ \forall l \neq j. \tag{10}$$

This result follows from the fact that as the population becomes more plastic as λ → 0 and $R(\lambda) \rightarrow SRS^T$, the canonical form of the fitness matrix is reduced to

$$\tilde{R}(0) = \begin{pmatrix} 0 & C_{12} & C_{13} \\ C_{21} & 0 & C_{23} \\ C_{31} & C_{32} & 0 \end{pmatrix},$$

where $C_{ij} = (\mathbf{s}_i - \mathbf{s}_j)^T R \mathbf{s}_j$ and $\mathbf{s}_i$ are the corresponding rows of *S*. If we think about $\mathbf{s}_j$'s as a mixed strategy that population use when vertex *j* is stable, we can interpret those conditions as stability requirements. That is, the plastic behaviour of strategy *j* has to be a better response to itself than both mixed profiles of strategies *i* and *k*. Hence, stability of the strategic choice of pure strategy *i* is determined by the stability of its mixed profile.

**Remark.** Note that for λ = 0, conditions (10) imply stability of vertex *j* for any number of strategies *n* and any game.

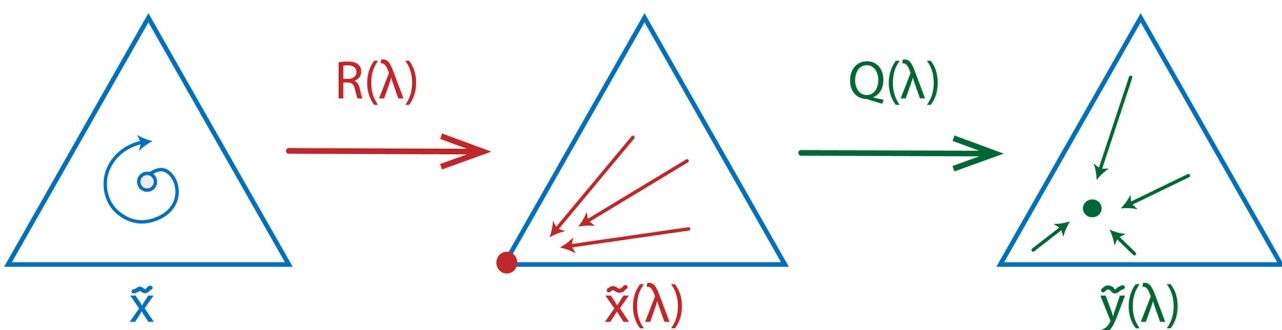

**Fig 5. A schematic representation of a possible influence of execution errors on the RPS dynamics.** The original game possesses an unstable equilibrium $\tilde{\mathbf{x}}$. Once the execution errors are introduced, for some $\lambda$ the game can obtain a stable equilibrium $\tilde{\mathbf{x}}(\lambda)$ represented by a vertex $i$. As the probability to play mixed strategies in this case is high, it keeps disturbing the strategy execution of the players choosing strategy $i$ according to Eq (7) resulting in an equilibrium $\tilde{\mathbf{y}}(\lambda)$ that is possibly in the interior.

Note that since the stable equilibrium is pure, it is a strict Nash equilibrium. Hence, the original replicator dynamics obtains an evolutionary stable point under the assumptions of our model. In fact, by Eq (7), according to the strategy execution of individuals, we obtain a stable point in the interior that corresponds to $\mathbf{s}_i$, where $i$ is a stable vertex. A schematic representation of such a transformation can be found in Fig 5.

As demonstrated in Examples 1 and 2, while $\lambda$ decreases from 1 to 0, the dynamics can experience several bifurcations where an equilibrium can emerge on one of the edges. An edge-equilibrium is characterised by exactly one of the components of the equilibrium being 0, that is, $\tilde{x}_i(\lambda) = 0$. Hence, this edge point is determined by the interaction between the two remaining strategies. However, for the version of an RPS game considered here, those equilibria will be mostly unstable (see S1 File for more details). Note that the point on the corresponding edge might exist before the interior reaches the boundary.

Overall, when strategies are initially equivalent in their fitness advantages in the non-plastic game, the asymmetry in matrix $Q(\lambda)$ introduces asymmetry in the game $\tilde{R}(\lambda)$. This in turn leads to the competition between pure strategies resulting in some stable fixed point of the dynamics. The outcome of the competition is determined by the interplay among mixed profiles of all three strategies. Specifically, the mixed profile of strategy $j$ has to be uninvadable by both populations consisting of individuals following strategies $i$ and $k$. This observation is different from the case of a uniform mixed strategies profile. That is, if in the former case, mixed profiles were likely to introduce a completely mixed equilibrium, in the case of asymmetric $S$, competition between the mixed strategies is important.

## Discussion

Much research has been devoted to describing behavioural mistakes of organisms and how those mistakes affect the outcome of the evolutionary competition. In addition, the RPS game itself received a lot of attention due to its ability to describe cyclic competitive interactions. However, such cycles are rarely observed in nature. We propose that behavioural heterogeneity or noise can induce stabilisation of communities driving them to evolutionary stable outcomes. Our model introduces behavioural mistakes in the context of a cyclic RPS game. Here, behavioural mistakes imply that individuals might execute a strategy different from the intended one. We encode all probabilities of mistakes in a matrix $Q(\lambda)$ and allow individuals to play either a mixed or pure strategy. The degree of plasticity is captured by the parameter $\lambda$ varying from 1 (no plasticity) to 0 (maximum plasticity).

We then explore the influence of the limiting distribution of mistakes captured in matrix $S$ on the evolution of social behaviour of species. Depending on the matrix $S$, different pure strategies might benefit from those mistakes. Such matrix captures mistake probabilities for the limiting case of $\lambda = 0$. We analyse the interplay of learning and fitness advantages and define conditions under which strategies can prevail. For example, in the case with completely random mistakes, the most beneficial strategy is the strategy with the highest relative fitness advantage (see Result 2). However, it does not change the outcome of the evolution since in this case it will be a completely mixed interior point.

One can also interpret our model as adaptation to new environmental conditions. Then, it is natural to expect that specific environments require different strategies to be adopted. For instance, in the case with an RPS game with the interior equilibrium $(^1/_3, ^1/_3, ^1/_3)$ and a general form of $S$, different strategies might become stable depending on their behavioural plasticity as their competence evolves (see Result 3). However, even if behavioural choice of organisms will evolve to a stable pure strategy, their executed strategy (for $\lambda \neq 1$) might differ from their actual type. Conversely, we will obtain a vector of mixed strategies given by Eq (7). Hence, $S$ can introduce stability in the game which might preserve all three strategies from extinction.

Interestingly, at $\lambda = 0$, strategies are leveraging the advantage they can gain from mistakes from maximum plasticity. For instance, in the case with a general form of limiting probability distribution, stability of a pure strategy is determined by its plastic response to itself (see Result 3). For a strategy to become stable, it is necessary to be uninvadable by the other two plastic strategies.

Overall, behavioural heterogeneity, captured through the execution noise, might help species to benefit from behavioural heterogeneity or plasticity. The ability of our model to induce a stable equilibrium in the unstable game might help in explaining why such unstable RPS dynamics are not observed in wild communities. That is, plasticity in behaviour might help to stabilise the evolutionary outcome and sometimes enable one of the strategies to become dominant.

## Supporting information

**S1 File. Mathematical appendix.** In this document we derive all results presented in the manuscript.
(PDF)

## Acknowledgments

Authors would like to thank Christian Hilbe and Martin Nowak for their inspiring and very helpful feedback on the manuscript.

## Author Contributions

**Conceptualization:** Maria Kleshnina, Jerzy A. Filar, Krishnendu Chatterjee.

**Formal analysis:** Maria Kleshnina, Sabrina S. Streipert.

**Funding acquisition:** Maria Kleshnina, Jerzy A. Filar, Krishnendu Chatterjee.

**Investigation:** Maria Kleshnina, Sabrina S. Streipert.

**Methodology:** Maria Kleshnina, Jerzy A. Filar.

**Project administration:** Maria Kleshnina.

**Supervision:** Jerzy A. Filar, Krishnendu Chatterjee.

**Visualization:** Maria Kleshnina, Sabrina S. Streipert.

**Writing – original draft:** Maria Kleshnina.

**Writing – review & editing:** Maria Kleshnina, Sabrina S. Streipert, Jerzy A. Filar, Krishnendu Chatterjee.

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
