## [Decision Letter · Decision Letter 0]

16 Feb 2021

Dear Dr Kleshnina,

Thank you very much for submitting your manuscript "Mistakes can stabilise the dynamics of rock-paper-scissors games" for consideration at PLOS Computational Biology. As with all papers reviewed by the journal, your manuscript was reviewed by members of the editorial board and by several independent reviewers. The reviewers appreciated the attention to an important topic. Based on the reviews, we are likely to accept this manuscript for publication, providing that you modify the manuscript according to the review recommendations.

Sincerely,

Attila Csikász-Nagy

Associate Editor

PLOS Computational Biology

Natalia Komarova

Deputy Editor

PLOS Computational Biology

[LINK]

Reviewer's Responses to Questions

**Comments to the Authors:**

Reviewer #1: In "Mistakes can stabilise the dynamics of rock-paper-scissors games" authors expand on the subject rock-paper-scissors dynamics, showing that mistakes in the strategy execution can stabilize different states. They consider an unstable version of rock-paper-scissors dynamics, such that allows individuals to make behavioural mistakes during the strategy execution. Research shows that such an assumption can break a cyclic relationship leading to a stable equilibrium emerging with only one strategy surviving. Different cases for the setup of mistakes are considered, both with interesting insights into the dynamics of the rock-paper-scissors dynamics.

The study of cyclic dominance models is an intensely investigated subject with obvious practical ramifications. Nonlinear science and methods of statistical physics have been used successfully and with much effect to outline many innovative ways on how such systems could be studied and solved, including many-species cyclic dominance models and the spontaneous emergence of cyclic dominance in related models. In this sense, the study addresses a relevant problem, and it also delivers results that will surely be of interest to the readership of PLOS Computational Biology.

The paper is well-written, comprehensive, and clear. I find it is among the finest papers that I have had the pleasure of reading in the recent past. The motivation behind the approach and the insights it affords towards improving spreading of communicable diseases is genius, and as such it will surely not fail to impress the diverse readership of PLOS Computational Biology. For these reasons, I warmly recommend publication subject only to a minor revision.

Namely, a recent work related to cyclic dominance is A novel route to cyclic dominance in voluntary social dilemmas, J. R. Soc. Interface 17, 20190789 (2020), which could be useful for a more up to date introduction. A good review is Cyclic dominance in evolutionary games: A review, J. R. Soc. Interface 11, 20140735 (2014).

It would also improve the paper if the figure captions would be made more self contained. In addition to what is shown for which parameter values, one could also consider a sentence or two saying what is the main message of each figure.

Apart from this, I am happy to congratulate the authors to an excellent contribution.

Reviewer #2: This is a very well written manuscript that can be accepted as is.

Only few minor comments are below

line 111 - explain why it is sufficient

Figure 2 - explain the coloring within the triangle

While I understand why the section "interpretation of lambda" is in the manuscript, it creates some confusion because authors talk about lambda "evolving" while the model assume lambda being constant. Perhaps there can be some discussion about possible extension of the model where even lambda undergoes some kind of a dynamics

**Have all data underlying the figures and results presented in the manuscript been provided?**

Reviewer #1: None

Reviewer #2: Yes

PLOS authors have the option to publish the peer review history of their article (what does this mean?). If published, this will include your full peer review and any attached files.

Reviewer #1: No

Reviewer #2: No

Figure Files:

Data Requirements:

Reproducibility:

References:

---

## [Editor Report · Decision Letter 1]

18 Mar 2021

Dear Dr Kleshnina,

We are pleased to inform you that your manuscript 'Mistakes can stabilise the dynamics of rock-paper-scissors games' has been provisionally accepted for publication in PLOS Computational Biology.

Best regards,

Attila Csikász-Nagy

Associate Editor

PLOS Computational Biology

Natalia Komarova

Deputy Editor

PLOS Computational Biology

---

## [Editor Report · Acceptance letter]

7 Apr 2021

PCOMPBIOL-D-20-02067R1 

Mistakes can stabilise the dynamics of rock-paper-scissors games

Dear Dr Kleshnina,

I am pleased to inform you that your manuscript has been formally accepted for publication in PLOS Computational Biology. Your manuscript is now with our production department and you will be notified of the publication date in due course.

With kind regards,

Alice Ellingham
